# Favorable Humoral Response to Third Dose of BNT162b2 in Patients Undergoing Hemodialysis

**DOI:** 10.3390/jcm11082090

**Published:** 2022-04-08

**Authors:** Mineaki Kitamura, Takahiro Takazono, Kosei Yamaguchi, Hideshi Tomura, Kazuko Yamamoto, Takashi Harada, Satoshi Funakoshi, Hiroshi Mukae, Tomoya Nishino

**Affiliations:** 1Nagasaki Renal Center, Nagasaki 850-0032, Japan; tugumasayamaguchi@gmail.com (K.Y.); ttoomm220011@nagasaki-u.ac.jp (H.T.); renaltharada@nagajin.jp (T.H.); satoshi2754@yahoo.co.jp (S.F.); 2Department of Nephrology, Nagasaki University Graduate School of Biomedical Sciences, Nagasaki 852-8523, Japan; tnishino@nagasaki-u.ac.jp; 3Department of Infectious Diseases, Nagasaki University Graduate School of Biomedical Sciences, Nagasaki 852-8523, Japan; takahiro-takazono@nagasaki-u.ac.jp (T.T.); kazukomd@nagasaki-u.ac.jp (K.Y.); 4Department of Respiratory Medicine, Nagasaki University Hospital, Nagasaki 852-8501, Japan; hmukae@nagasaki-u.ac.jp; 5Department of Respiratory Medicine, Nagasaki University Graduate School of Biomedical Sciences, Nagasaki 852-8523, Japan

**Keywords:** coronavirus disease 2019, hemodialysis, humoral response, immunogenicity, severe acute respiratory syndrome coronavirus 2, vaccination, third dose

## Abstract

Patients undergoing hemodialysis are known to exhibit low humoral responses to vaccines against severe acute respiratory syndrome coronavirus 2. In this study, we aimed to elucidate the humoral response to the third dose of BNT162b2 (Pfizer) in patients undergoing hemodialysis. We included 279 patients undergoing hemodialysis (69 ± 11 years, 65% male, median dialysis vintage: 69 months) and 189 healthcare workers (45 ± 13 years, 30% male) who received the third dose of BNT162b2. Anti-spike immunoglobulin G (anti-S IgG) antibody levels were measured 3–4.5 months after the second dose and 3 weeks after the third dose and were compared. Despite a significant difference in anti-S IgG antibody levels after the second dose between the two groups (patients: median 215 U/mL and healthcare workers: median 589 U/mL; *p* < 0.001), no significant difference in anti-S IgG antibody levels after the third dose was observed (patients: median 19,000 U/mL, healthcare workers: median 21,000 U/mL). Except for dialysis vintage (ρ = 0.209, *p* < 0.001), no other factors correlated with anti-S IgG antibody levels after the third vaccine dose in patients undergoing hemodialysis. Therefore, a favorable response to the third dose of BNT162b2 was observed in patients undergoing hemodialysis, irrespective of their backgrounds.

## 1. Introduction

Effective mRNA vaccines against severe acute respiratory syndrome coronavirus 2 (SARS-CoV-2), such as BNT162b2 (Pfizer) and m-RNA1273 (Moderna), are now widely available [1,2]. These vaccines prevent the contraction of coronavirus disease 2019 (COVID-19), and even if the disease is contracted, fully vaccinated patients exhibit mild symptoms. However, anti-spike immunoglobulin G (anti-S IgG) antibodies decrease rapidly, and a third dose of the vaccine is recommended several months after the second dose [3]. Moreover, patients who do not show sufficient anti-S IgG antibody levels following the administration of the second dose are recommended to receive the third dose regardless of vaccine intervals in some countries [4]. It is well-known that elderly people, patients on immunosuppression therapy, organ transplant recipients, and patients with diabetes show low-immune reactivity to vaccinations [5,6]. Additionally, patients with renal failure also tend to show low reactivity to vaccinations [7,8].

Previous studies have shown that patients undergoing hemodialysis (HD) are more susceptible to the contraction of COVID-19, and their mortality rate (approximately 20%) is higher than that seen in a healthy population [9,10,11]. As patients undergoing HD show relatively low humoral response to vaccinations, there are concerns that the efficacy of the vaccination against SARS-CoV-2 is lower in patients undergoing HD than in healthy populations. The anti-S IgG antibody levels after the second dose of mRNA vaccines among patients undergoing HD are lower [12,13] and wane earlier compared to those in healthy populations [11,14,15,16]. A third dose of mRNA vaccination is recommended for patients undergoing HD; however, there are limited reports that elucidate the effect of the third dose of mRNA vaccines on anti-S IgG antibody levels in patients undergoing HD [17,18,19,20,21]. Of these, only one report compared the humoral response between patients undergoing HD and the control group, but the clinical features of the patients were not described [19]. As a result of improvements in healthcare practices and an increase in the average age of society, the mean age of patients undergoing HD has increased worldwide, and the humoral response is inversely correlated with age in the patients undergoing HD [12]. Because of the higher risk of contracting COVID-19 in facilities for HD [22,23], effective anti-S IgG antibody levels should be present in all patients undergoing HD.

This study aimed to clarify and compare the humoral response in patients undergoing HD and in healthy populations to the third dose of the mRNA vaccine against SARS-CoV-2. In addition, the association between clinical features and the development of anti-S IgG antibodies after the third dose of BNT162b2 in patients undergoing HD was explored.

## 2. Materials and Methods

### 2.1. Study Participants

Patients undergoing HD who were administered a third dose of BNT162b2 (30 μg intramuscular injection) at Nagasaki Renal Center and Nagasaki Renal Clinic from January 2022 to February 2022 were included in the study. Healthcare workers who received the third dose of BNT162b2 at our facilities were also included as a control group. Patients and healthcare workers who had a prior history of contracting COVID-19 were excluded from this study. Patients who did not undergo routine blood exams in January 2022 were also excluded from this study.

### 2.2. Anti-Spike Protein Measurement

Due to the concern of COVID-19 outbreaks at our facility, anti-S IgG antibody levels were measured in patients who received BNT162b2 at the two facilities in October 2021. This timing corresponded to 4.5 months after the second dose of BNT162b2 for healthcare workers and 3 months for patients undergoing HD. The third dose of BNT162b2 was given to the healthcare workers in December 2021 and to the patients undergoing HD in January 2022. To confirm the efficacy of the third dose, anti-S IgG antibody levels were measured 3 weeks after the third dose.

Anti-S IgG antibody levels were measured using the Elecsys^®^ Anti-SARS-CoV-2 immunoassay (Roche Diagnostics International Ltd., Risch-Rotkreu, Switzerland). According to the manufacturer’s instructions, the cut-off level for seronegativity was set at 0.8 U/mL. A patient was classified as a low responder if their anti-S IgG antibody levels were <80 U/mL and as a responder if their anti-S IgG antibody levels were ≥80 U/mL, as described in a previous study [17]. Nucleocapsid (N) antibody levels were measured using the Elecsys^®^ assay to exclude subjects with a prior history of contracting COVID-19 (cut-off index < 1.0 interpreted as non-reactive (negative)). We collected the background data of patients from their medical records. The result of the blood examination was obtained from routine clinical practice in January 2022. The background data of the healthcare workers were collected from their latest medical health checks. 

### 2.3. Statistical Analysis

Continuous values are shown in means ± standard deviations or medians with their respective interquartile ranges (IQRs) if values are not normally distributed. Categorical variables are shown as numbers (%). Student’s *t*-test, Wilcoxon rank-sum, and chi-squared tests were used to evaluate differences between the two study groups. Spearman’s correlation analysis was used to calculate correlations between two continuous variables, and “ρ” was used as a correlation coefficient. Body mass index was calculated using the patients’ dry weight and height. A *p*-value of <0.05 was considered statistically significant. Statistical analyses were conducted using the JMP Pro 15.0.0 (3903308) (SAS Institute Inc., Cary, NC, USA). 

## 3. Results

Among the patients undergoing HD, five patients contracted COVID-19 before receiving the third dose, and six patients contracted COVID-19 within 3 weeks after the third dose of BNT162b2. Three healthcare workers had a prior history of contracting COVID-19 at the time of the third dose. Incidentally, three patients declined to receive the third dose of BNT162b2 and did not provide consent to have their anti-S IgG levels measured. After excluding those patients and healthcare workers, a total of 279 patients and 189 healthcare workers were included in this study. The mean age of the patients undergoing HD was 69 ±11 years, and 182 patients (65%) were male. In contrast, the mean age of the healthcare workers was 45 ± 13 years, and 57 healthcare workers (30%) were male. The clinical features of the patients and healthcare workers are shown in Table 1. 

The median anti-S IgG antibody levels of patients undergoing HD 3 months after the second dose was 215 (IQR: 103–387) U/mL (Table 2). All participants were found to be seropositive. Although the anti-S IgG antibody level was less than 80 U/mL in one healthcare worker (0.5%), anti-S IgG antibody levels of 52 patients (18.6%) were less than 80 U/mL at this point. There were no significant differences between patient backgrounds of low-responders and responders undergoing HD (Table 3). On the contrary, the median anti-S IgG antibody level of healthcare workers was 589 (IQR: 396–853) U/mL (*p* < 0.001). All healthcare workers maintained sufficient anti-S IgG antibody levels even 4.5 months after the second dose (Figure 1). 

The third dose was administered to both groups approximately 7 months after the second dose. There was no significant difference in anti-S IgG antibody levels between the patients undergoing HD (19,000 U/mL (IQR: 12,000–61,000) and healthcare workers (21,000 U/mL (IQR: 15,000–36,500)) (Table 2). All participants showed an increase in their anti-S IgG antibodies to suitable levels (>80 U/mL). The lowest anti-S IgG antibody levels were 206 U/mL and 2280 U/mL in patients undergoing HD and healthcare workers, respectively (Figure 1). The increasing rate from the measurement in October 2021 showed that the response of patients undergoing HD to the third dose of BNT162b2 was more than three times stronger than in healthcare workers (*p* < 0.001) (Table 2). Although the anti-S IgG antibody levels in the low responder in patients undergoing HD were lower than those in other patients, the rate of increase in low responders was about two times higher than that in other patients undergoing HD (Table 3). 

Spearman’s analyses showed that HD vintage was positively correlated with the anti-S IgG antibody levels 3 weeks after the third dose in the patients undergoing HD (*p* < 0.001). White blood cell and body mass index in patients undergoing HD were correlated with the increasing rate ((anti-S IgG antibody levels 3 weeks after the third dose)/(anti-S IgG antibody levels 3 months after the second dose)). However, there was no significant correlation between age and the increase in anti-S IgG antibody levels (Table 4).

Age was negatively correlated with anti-S IgG antibody levels 3 weeks after the third dose in healthcare workers (*p* = 0.002). Inversely, the increasing rate ((anti-S IgG antibody levels 3 weeks after the third dose)/(anti-S IgG antibody levels 4.5 months after the second dose)) was positively correlated with age in the healthcare workers (*p* = 0.004) (Table 5). 

The correlation between continuous parameters and anti-S IgG antibody levels in October 2021 is shown in Appendix A. 

There was no significant difference in anti-S IgG antibody levels between males and females during the observation period in patients undergoing HD (Appendix A). Although female healthcare workers showed significantly higher anti-S IgG antibody levels in October 2021 (*p* = 0.02), the third dose of BNT162b2 resulted in equal anti-S IgG antibody levels in male and female healthcare workers. Therefore, there was no significant difference in antibody levels between male and female healthcare workers (*p* = 0.68) (Appendix A). History of diabetes did not affect anti-S IgG antibody levels in both the patients undergoing HD and healthcare workers (Appendix A).

Among patients undergoing HD, eight patients received glucocorticoids, and their anti-S spike protein levels (median 21,500 U/mL) were higher than that of patients who had not received glucocorticoids (median 19,000 U/mL). However, there was no significant difference between the two groups (Appendix A).

No severe adverse events against the third dose of BNT162b2, such as anaphylaxis, were observed in the patients or healthcare workers.

## 4. Discussion

In this study, we compared the effect of the third dose of BNT162b2 in patients undergoing HD and healthcare workers (control group). To the best of our knowledge, the number of patients and controls in this study was larger than those seen in previous reports [17,18,19,20,21]. Even though anti-S IgG antibody levels in patients undergoing HD 3 months after the second dose were lower than those of healthcare workers 4.5 months after the second dose, there was no significant difference in the anti-S IgG antibody levels between the patients undergoing HD and healthcare workers after the third dose. The anti-S IgG antibody levels in the patients undergoing HD, especially in low responders, increased considerably after the third dose of BNT162b2. All participants in this study responded to the third dose and developed sufficient levels of anti-S IgG antibodies. Our results suggested that the third dose of BNT162b2 is crucial for patients undergoing HD and is effective irrespective of patients’ background. 

There have been several reports on the effectiveness of the third dose of BNT162b2 in patients undergoing HD [17,18,19,20,21]. Bensouna et al. revealed that patients undergoing HD showed median anti-S IgG antibody levels of 7554 U/mL after the third dose of BNT162b2, even though the median after the second dose was 284 U/mL [18] (titers in that study were measured using a similar system to ours). This result indicates that the third dose allows patients to develop more than 25 times greater antibody levels after the second dose. Moreover, all patients produced sufficient levels of anti-S IgG antibodies [18]. Similar trends were observed in this study; however, the efficacy of the third dose was higher in this study. Although we did not evaluate the anti-S IgG antibody levels just before the third dose, the rate of increase in antibody levels in the patients undergoing HD would be higher. There are no plausible speculations regarding this fact, but some differences in the characteristics of the patients, such as race, might be associated with this phenomenon. Another report from Israel showed that the anti-S IgG antibody levels after the third dose of BNT162b2 in patients undergoing HD matched that of the healthy population [19]. Similarly, in this study, the patients undergoing HD exhibited increased anti-S IgG antibody levels due to their favorable response to the third dose. 

The proportion of elderly people in the population has increased in the western world, and they are more susceptible to COVID-19 infection and face a higher risk of mortality [24]. Although full vaccination for COVID-19 is mandatory in elderly people, they are often immunodeficient, and their humoral response to vaccination is weaker than that observed in young people [12]. Furthermore, their anti-S IgG antibody levels wane earlier [25]. Renal failure is more prevalent in the elderly population, and the mean age of patients undergoing HD is increasing in almost all countries [24]. Therefore, monitoring anti-S IgG antibody levels in elderly patients undergoing HD can aid in devising a suitable vaccination plan for them [26]. Although there was a significant correlation between age and anti-S IgG antibody levels in the healthcare workers, this trend was not observed in the patients undergoing HD in this study. This might be because of the narrow age range of the patients surveyed in this study compared to that in the previous study that showed the age-dependent humoral response in patients undergoing HD [12]. Even elderly patients undergoing HD achieved a good humoral response to the third dose of BNT162b2 in this study. 

Patients undergoing HD are known to be an extremely immunodeficient population at high risk of contracting severe COVID-19 [9,10,11]. Several mechanisms account for their vulnerability to infectious diseases, and uremic toxins and pro-inflammatory cytokines play an important role in their increased vulnerability [27]. Decreased humoral cellular responses are associated with infections, but adaptive immune system disorder is hindered by uremia [27]. Previous studies elucidated that the rate of achieving seroconversion was gradually increased after COVID-19 recovery [28] or after receiving two doses of the mRNA vaccine for SARS-CoV-2 [29]. This may be because immune systems in patients undergoing HD are interfered with by uremic toxins [27,30]. Dialysis vintage was positively associated with anti-S IgG antibody levels 3 months after the second dose and 3 weeks after the third dose in this study. As immune disorder in patients undergoing HD is multifactorial, it is difficult to deduce the reason why patients with long HD vintage had higher anti-S IgG antibody levels. We speculate that patients with long-term dialysis vintage may have few complications because patients undergoing HD with multiple comorbidities have a poor life prognosis [31]. As the immune system is an important factor in longevity [32], patients undergoing HD who show a good immune response can survive over a long-term period without contracting infectious diseases. 

Patients undergoing HD have to visit HD facilities weekly thrice, regardless of COVID-19. Therefore, as a result of increased exposure, their risk of contracting COVID-19 is higher [22,23]. Similarly, healthcare providers working at HD facilities face a high risk of contracting COVID-19 due to the nature of their work. Patients undergoing HD, as well as healthcare workers, cannot reasonably maintain suitable social distancing practices to avoid contracting COVID-19 [33]. To decrease the risk, they need to pay careful attention to sanitation practices and be fully vaccinated. 

After the emergence of the omicron variant of SARS-CoV-2, the number of patients who tested positive for COVID-19 increased worldwide [34]. The omicron variant has high transmissibility because of some deletions and more than 30 mutations [30]. In addition, the latest reports revealed that the efficacy of the BNT162b2 vaccine against this variant is limited (70%) compared to that of other major variants, such as the delta variant (93%) [35]. Six patients undergoing HD at our facilities contracted COVID-19 even after receiving the third dose. The omicron variant was prevalent in Japan during the study period, and these six patients were supposed to have contracted this variant. Fortunately, their symptoms were very mild and improved in a short period. In contrast, no healthcare workers contracted COVID-19 after the third dose, suggesting that the efficacy of the third dose of BNT162b2 cannot be judged only by the humoral response. Nonetheless, the neutralization effect of the third dose of BNT162b2 against the omicron variant is superior to that of the second dose [36]. Although the efficacy of the third dose of BNT162b2 in patients undergoing HD has not been elucidated yet, a third dose of the vaccine would be effective in preventing outbreaks among patients undergoing HD. 

This study, however, has several limitations. First, we evaluated anti-S IgG antibody levels in both HD patients and healthcare workers at our facilities but only measured them at two points in the patients included in this study. As there was a time lag in receiving the second dose of BNT162b2 between patients undergoing HD and healthcare workers, several discrepancies existed between the two groups in the measurement performed in October 2021. Consequently, we could not show the result of the time course of vaccination. Additionally, we defined non-responder using the same standard of a previous study [17]; however, our measurements were done 3–4.5 months after the second dose, and there could be discrepancies from previous investigations. Second, this study was conducted at two facilities, and the patient background might be different from those in other facilities. Third, as described above, the efficacy of BNT162b2 is not defined only through anti-S IgG antibody levels. For example, the cellular response against SARS-CoV-2 is crucial as well as the humoral response. Moreover, even if patients were fully vaccinated with a high titer of anti-SARS-CoV-2 protein, the efficacy of BNT162b2 is yet to be fully proven against the latest omicron variant.

## 5. Conclusions

The patients undergoing HD acquired the levels of anti-S IgG antibody levels matching those observed in healthcare workers after being administered the third dose of BNT162b2. As the patients undergoing HD at healthcare facilities face a higher risk of contracting COVID-19, the third dose of BNT162b2 should be considered to reduce mortality and prevent COVID-19 outbreaks at HD facilities. Future studies are needed to evaluate the efficacy of the third dose of BNT162b2 in patients undergoing HD in real-world conditions. 

## Figures and Tables

**Figure 1 jcm-11-02090-f001:**
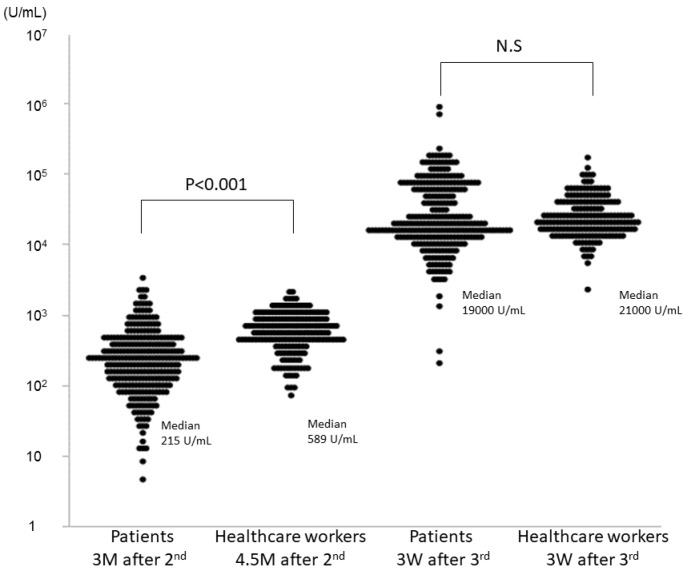
Levels of anti-spike protein immunoglobulin G (IgG) antibody against SARS-CoV-2, 3 months after the second dose of BNT162b2 in patients undergoing hemodialysis and 4.5 months after the second dose of BNT162b2 in the healthcare workers. Levels of IgG antibodies 3 weeks after the third dose of BNT162b2 are shown in both groups.

**Table 1 jcm-11-02090-t001:** Characteristics of the patients on hemodialysis and the healthcare workers.

	Patients on HD (*n* = 279)	Healthcare Workers (*n* = 189)	*p*-Value
**Age (years)**	69 ± 11	45 ± 13	<0.001
**Sex (male) (%)**	65	30	<0.001
**BMI (kg/m^2^)**	21.3 ± 3.7	21.8 ± 4.2	0.28
**Dialysis vintage (months)**	69 (34–141)	NA	
**Diabetes mellitus (%)**	44	5	<0.001
**Hypertension (%)**	91	6	<0.001
**History of ischemic heart diseases (%)**	52	0	<0.001
**History of stroke (%)**	22	0	<0.001
**Mean KT/V**	1.49 ± 0.30	NA	
**White blood cell count (/μL)**	6275 ± 2179	6224 ± 1367	0.80

HD—hemodialysis; BMI—body mass index; NA—not available. Mean ± standard deviation, or median (interquartile range). Student’s *t*-test or Wilcoxon rank-sum and chi-squared tests were used.

**Table 2 jcm-11-02090-t002:** Humoral response to BNT162b2 vaccination in terms of anti-spike immunoglobulin G antibody levels.

	Patients on HD (*n* = 279)	Healthcare Workers (*n* = 189)	*p*-Value
** Measurement in October 2021 **			
**Days since the second dose of BNT162b2**	91 ± 13	138 ± 4	<0.001
**(1) anti-S IgG antibody levels at the follow-up of the second dose (U/mL)**	215 (103–387)	589 (396–853)	<0.001
**Days between the second dose and the third dose of BNT162b2 (days)**	202 ± 11	219 ± 5	<0.001
** Measurement 3 weeks after the third dose **			
**Days since the third dose**	21 ± 1	20 ± 3	<0.001
**(2) anti-S IgG antibody levels at the follow-up of the third dose (U/mL)**	19,000 (12,000–61,000)	21,000 (15,000–36,500)	0.90
**Increasing rate calculating from (2)/(1)**	129 (59–216)	37 (25–69)	<0.001

HD—hemodialysis; anti-S IgG—anti-spike immunoglobulin G. Mean ± standard deviation, or median (interquartile range). Student’s *t*-tests or Wilcoxon rank-sum tests were used.

**Table 3 jcm-11-02090-t003:** Difference between low responder and responder in patients on maintenance hemodialysis.

	Low-Responder in Patients on HD (*n* = 51)	Responder in Patients on HD (*n* = 228)	*p*-Value
**Age (years)**	69 ± 11	69 ± 12	0.87
**Sex (male) (%)**	63	66	0.68
**BMI (kg/m^2^)**	22.5 ± 4.1	21.7 ± 4.2	0.08
**Dialysis vintage (months)**	60 (35–156)	75 (33–141)	0.50
**Diabetes mellitus (%)**	33	46	0.09
**Mean KT/V**	1.43 ± 0.25	1.50 ± 0.30	0.16
**White blood cell count (/μL)**	6815 ± 2272	6154 ± 2144	0.0502
**Hemoglobin (g/dL)**	10.8 ± 1.2	10.8 ± 1.0	0.96
**Albumin (g/dL)**	3.5 ± 0.4	3.6 ± 0.4	0.14
**(1) anti-S IgG antibody levels at the follow-up of the second dose (U/mL)**	46 (33–67)	246 (155–435)	<0.001
**(2) anti-S IgG antibody levels at the follow-up of the third dose (U/mL)**	9900 (4560–17,000)	21,000 (14,000–66,750)	<0.001
**Increasing rate calculating from (2)/(1)**	212 (124–613)	105 (56–193)	<0.001

HD—hemodialysis; BMI—body mass index; NA—not available. Mean ± standard deviation, or median (interquartile range). Student’s *t*-test or Wilcoxon rank-sum and chi-squared tests were used.

**Table 4 jcm-11-02090-t004:** Correlation between continuous parameters and anti-spike immunoglobulin G antibody levels 3 weeks after the third dose of BNT162b2.

	Patients on HD		Healthcare Workers	
	ρ	*p*-Value	ρ	*p*-Value
**Age**	−0.111	0.06	−0.227	0.002
**Dialysis vintage (months)**	0.209	<0.001	NA	
**Body mass index**	0.115	0.06	0.208	0.01
**White blood cell count**	0.094	0.12	0.080	0.35
**Hemoglobin**	−0.115	0.05	0.067	0.43
**Albumin**	0.001	0.99	NA	
**KT/V**	0.097	0.11	NA	

HD—hemodialysis; NA—not available. Spearman analysis was used.

**Table 5 jcm-11-02090-t005:** Correlation between continuous parameters and rate of increase in anti-spike immunoglobulin G antibody levels from October 2021 3 weeks after the third dose of BNT162b2.

	Patients on HD		Healthcare Workers	
	ρ	*p*-Value	ρ	*p*-Value
**Age**	−0.042	0.49	0.210	0.004
**Dialysis vintage (months)**	0.084	0.16	NA	
**Body mass index**	0.210	<0.001	0.144	0.09
**White blood cell count**	0.128	0.03	0.024	0.78
**Hemoglobin**	−0.078	0.19	0.076	0.37
**Albumin**	−0.112	0.06	NA	
**KT/V**	−0.012	0.84	NA	

HD—hemodialysis; NA—not available. Spearman analysis was used.

## Data Availability

The data underlying this article will be shared upon reasonable request from the corresponding author.

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
