# Peer review of "Favorable Humoral Response to Third Dose of BNT162b2 in Patients Undergoing Hemodialysis"

_jcm, 2022, doi:10.3390/jcm11082090_

Round 1

Author Response

Reviewer 1.

Response: We would like to thank Reviewer 1 for the time and effort taken in reviewing our manuscript and for providing valuable comments and suggestions which have helped us considerably improve our manuscript.

The study “Favorable humoral response to a third dose of BNT162b2 in patients undergoing hemodialysis” by Kitamura et. al., provides important evidence pertaining to probable benefits of a third dose of the BNT162b2 vaccine in immunocompromised patients, specifically those undergoing hemodialysis (HD).

The study included a large number of patients and healthy volunteers allowing for reliable interpretation of the presented results. The authors evaluated the antibody response in 279 patients undergoing HD (69±11 years old, 65% male, median dialysis vintage: 69 months) and 189 healthcare workers (45±13 years old, 30% male) who received the third dose of BNT162b2. They measured anti-spike antibody levels at two time-points and compared the responses between the 2 groups. The results are easy to follow and have been well described with appropriate details pertaining to cohort characteristics and statistical testing. The discussion is also well-presented including limitations of the current study. Although quite informative, as the authors have acknowledged, the significance of this study does not reach its full potential due to reliance on a single assay (standard anti-spike antibody measurement). Additional plasma/serum measurements (neutralization/anti-RBD antibody titers) will significantly add to the current data. These assays can further inform if there are differences in the quality of response between the two groups in terms of the targeted epitopes etc.? Do the authors have PBMCs banked from these donors to perform cellular assays?

Response: Thank you for your constructive and informative comments. As our study was based on clinical practice, the plasma/serum measurements were not conducted. We did not store peripheral blood mononuclear cells, either. However, we are going to investigate the neutralization ability of antibodies obtained from some participants in concert with a medical laboratory. The data have not been obtained yet, and the number of cases is limited; therefore, we are unable to show the result.  

Another limitation of the study is that the post third-dose measurement is <1 month after vaccination. Their data showing differences between HD patients and healthcare workers post second-dose is at a much later time-point (>3 months) and probably is also indicative of faster decay of antibodies in serum of HD patients. It will be helpful if the authors can provide data and/or discuss how the benefit of a third dose in HD patients can be limited due to faster decay.

Response: We agree with your opinion. We are going to measure anti-S IgG antibody levels 4 months after the third dose, and the result is not available now. We anticipate that anti-S IgG antibody levels in patients undergoing HD will be low compared to the healthy control.  

Reviewer 2 Report

The work is of considerable interest,
I would have some points to report:
1) the number of people tested is not written in the text, so include the number of participants in the Study participants paragraph.
2) it would be useful for completeness of the results that the authors include the test of neutralization 

Author Response

Reviewer 2.

Response: We would like to thank Reviewer 2 for the time and effort taken in reviewing our manuscript and for providing valuable comments and suggestions which have helped us in considerably improving our manuscript.

The work is of considerable interest,

I would have some points to report:

1) the number of people tested is not written in the text, so include the number of participants in the Study participants paragraph.

Response: Thank you for your comment. We measured anti-S IgG antibody levels in patients who received the third dose of BNT162b2. A total of 279 patients and 189 healthcare workers were included in this study. Among our patients, three refused to receive the third dose of BNT162b2. Furthermore, they declined to have their anti-S IgG levels measured. Therefore, we did not measure their anti-S IgG antibody levels. We have stated the number of patients and healthcare workers involved in the study (Line 115–121).

2) it would be useful for completeness of the results that the authors include the test of neutralization

Response: Our measurement system here does not evaluate neutralization. However, we have investigated the neutralization ability of antibodies obtained from some participants in concert with a medical laboratory. The data have not been obtained yet, and the number of cases is limited; therefore, we are unable to show the result yet. We thank you for your valuable opinion.

Reviewer 3 Report

The study is timely and design is adequate, also the number of subjects enrolled is quite high. Also it is focused on probably naive to CoV-2 subjects.

It is not specified the proportion of patients among HD group with IMID, and the proportion of subjects still on glucococorticoids or immunosuppressants/DMARS. This should also be analyzed with appropriate statistical methods at the timepoints used in the study.

Another point is that with the anti-spike method used in the study, they classify the subjects in low-responders, and normal responders. It's unclear this kind of distinction. If the cut-off for positvity of the method is 80 U/ml (e.g. after second dose or after third dose), those are non-responders insted of low-responders. It is also linked to analytical properties of the CLIA(?) method used here. The point for those patients is the lack of response, more than "low response". Clear rates of responders and non-responders (defined by ab seroconversion), should be given. Antibody titres should be calculated only on the responders.

Round 2

Reviewer 2 Report

I accept this article

Reviewer 3 Report

the authors improved the manuscript and clarity of methods and data